

# Soil carbon loss in warmed subarctic grasslands is rapid and restricted to topsoil

Niel Verbrigghe[1], Niki I. W. Leblans[1,2], Bjarni D. Sigurdsson[3], Sara Vicca[1], Chao Fang[1,4,5], Lucia Fuchslueger[1,6], Jennifer L. Soong[1,7], James T. Weedon[8], Christopher Poeplau[9], Cristina Ariza-Carricondo[1], Michael Bahn[10], Bertrand Guenet[11], Per Gundersen[12], Gunnhildur E. Gunnarsdóttir[13], Thomas Kätterer[14], Zhanfeng Liu[15], Marja Maljanen[16], Sara Marañón-Jiménez[17,18], Kathiravan Meeran[10], Edda S. Oddsdóttir[19], Ivika Ostonen[20], Josep Peñuelas[17,18], Andreas Richter[6,21], Jordi Sardans[17,18], Páll Sigurðsson[3], Margaret S. Torn[22], Peter M. Van Bodegom[23], Erik Verbruggen[1], Tom W. N. Walker[24], Håkan Wallander[25], and Ivan A. Janssens[1]

[1]Research Group Plants and Ecosystems, University of Antwerp, Antwerp, Belgium.
[2]Climate Impacts Research Centre, Umeå University, Umeå, Sweden.
[3]Agricultural University of Iceland, Hvanneyri, Borgarnes, Iceland.
[4]Institute of Ecology, School of Applied Meteorology, Nanjing University of Information Science and Technology, Nanjing, China.
[5]State Key Laboratory of Grassland Agro-ecosystems, Institute of Arid Agroecology, School of Life Sciences, Lanzhou University, Lanzhou, China.
[6]Centre for Microbiology and Environmental Systems Science, University of Vienna, Vienna, Austria.
[7]Soil and Crop Sciences Department, Colorado State University, Fort Collins, Colorado, USA.
[8]Systems Ecology, Department of Ecological Science, Vrije Universiteit Amsterdam, Amsterdam, The Netherlands.
[9]Thünen Institute of Climate-Smart Agriculture, Braunschweig, Germany.
[10]Department of Ecology, University of Innsbruck, Innsbruck, Austria.
[11]Laboratoire de Géologie, École normale supérieure/CNRS, PSL Research University, Paris, France.
[12]Department of Geosciences and Natural Resource Management, University of Copenhagen, Frederiksberg C, Denmark.
[13]Soil Conservation Service of Iceland, Gunnarsholt, Hella, Iceland.
[14]Department of Ecology, Swedish University of Agricultural Sciences, Uppsala, Sweden.
[15]Key Laboratory of Vegetation Restoration and Management of Degraded Ecosystems & CAS Engineering Laboratory for Vegetation Ecosystem Restoration on Islands and Coastal Zones, South China Botanical Garden, Chinese Academy of Sciences, Guangzhou, China.
[16]Department of Environmental and Biological Sciences, University of Eastern Finland, Kuopio, Finland.
[17]CREAF, Cerdanyola del Vallès, Barcelona, Catalonia, Spain.
[18]CSIC, Global Ecology Unit CREAF-CSIC-UAB, Bellaterra, Barcelona, Spain.
[19]Icelandic Forest Research, Mógilsá, Reykjavík, Iceland.
[20]Institute of Ecology and Earth Sciences, University of Tartu, Tartu, Estonia
[21]International Institute for Applied Systems Analysis (IIASA), Laxenburg, Austria.
[22]Climate and Ecosystem Sciences Division, Berkeley Lab, Berkeley, CA, USA.
[23]Environmental Biology Department, Institute of Environmental Sciences, CML, Leiden University, Leiden, The Netherlands.
[24]Department of Environmental Systems Science, ETH Zürich, Zürich, Switzerland.
[25]MEMEG, Department of Biology, Lund University, Lund, Sweden.

**Correspondence:** Niel Verbrigghe (Niel.Verbrigghe@UAntwerpen.be)





**Abstract.** Global warming may lead to carbon transfers from soils to the atmosphere, yet this positive feedback to the climate system remains highly uncertain, especially in subsoils (Ilyina and Friedlingstein, 2016; Shi et al., 2018). Using natural geothermal soil warming gradients of up to +6.4 °C in subarctic grasslands (Sigurdsson et al., 2016), we show that soil organic carbon (SOC) stocks decline strongly and linearly with warming ($-2.8$ ton ha$^{-1}$ °C$^{-1}$). Comparison of SOC stock changes

following medium-term (5 and 10 years) and long-term (>50 years) warming revealed that all SOC loss occurred within the first five years of warming, after which continued warming no longer reduced SOC stocks. This rapid equilibration of SOC observed in Andosol suggests a critical role for ecosystem adaptations to warming and could imply short-lived soil carbon-climate feedbacks. Our data further revealed that the soil C loss occurred in all aggregate size fractions, and that SOC losses only occurred in topsoil (0-10 cm). SOC stocks in subsoil (10-30 cm), where plant roots were absent, remained unaltered, even

after >50 years of warming. The observed depth-dependent warming responses indicate that explicit vertical resolution is a prerequisite for global models to accurately project future SOC stocks for this soil type and should be investigated for soils with other mineralogies.

## 1  Introduction

Soils store more carbon (C) than the atmosphere and vegetation biomass combined (Batjes, 2016; Scharlemann et al., 2014).

Global warming has been hypothesised to lead to increased soil CO$_2$ emissions that may lead to large reductions in soil organic carbon (SOC) stocks, constituting a positive feedback to the climate system (Davidson and Janssens, 2006; Jenkinson et al., 1991). The strength and even sign of this carbon cycle-climate feedback are, however, highly uncertain (Crowther et al., 2016; Todd-Brown et al., 2018; van Gestel et al., 2018). Accordingly, the World Climate Research Programme has acknowledged it as one of the "Grand Challenges" of climate research (Ilyina and Friedlingstein, 2016).

*In situ* soil warming studies provide ideal tools to study the response of soil SOC stocks to warming (Batjes, 2016), yet challenges remain great. First, ecosystem responses to warming may take decades to stabilise (Walker et al., 2020; Melillo et al., 2017), implying that extrapolations of responses from-, or model parametrisation based on short-term experiments may lead to erroneous estimation of the future evolution of SOC stocks. Second, SOC stock changes are rarely studied in subsoils. The high cost and labour requirements of SOC research, combined with the fact that most biological activity and

SOC mineralisation occur in topsoils, explains why soil biology and ecology, including SOC cycling, are rarely studied below a depth of 20-30 cm (Yost and Hartemink, 2020). This is a major issue, because more SOC is stored below this threshold than above (Shi et al., 2020) and therefore the carbon cycle-climate feedback does not stop at 20-30 cm depth. Unfortunately, the very few soil warming experiments that also warmed subsoils and quantified SOC stock changes yielded very different warming responses, ranging from declining to increasing subsoil SOC stocks (Soong et al., 2020a; Hanson et al., 2020).

To address both these challenges, we determined SOC stock changes along natural geothermal gradients at the ForHot research site in Iceland (Sigurdsson et al., 2016) encompassing the full warming range projected for Northern regions (up to +6.4 °C), throughout the topsoil (0-10 cm) and the subsoil (10-30 cm). We compared topsoil and subsoil SOC dynamics along replicate warming gradients exposed to medium-term (5 and 10 years) and long-term (>50 years, but possibly centuries)





warming by sampling permanent study plots twice in a six-year period (2013 and 2018). This enabled us to characterise the

magnitude, shape and temporal dynamics of the temperature response of SOC stocks in these northern, non-permafrost, soils. Next to measuring SOC stocks, we gathered data about soil aggregates, carbon inputs to the soil by plants and arbuscular mycorrhizal fungi and carbon flux from topsoil to subsoil. This allowed us to elaborate about the possible mechanisms behind SOC stock changes along the warming gradient.

The recently warmed grassland we investigated at the ForHot site has been warmed since 2008, when a major earthquake

shifted geothermal systems to previously unwarmed soils, causing increased temperature in the soil above by radiative heating (Halldórsson and Sigbjörnsson, 2009; O'Gorman et al., 2014). In contrast, the long-term warmed grassland had been warmed for at least 45 years at the time of the earthquake in 2008 (Sigurdsson et al., 2016). The soil type on both study sites is Andosol, and they are covered by the same grassland type (Sigurdsson et al., 2016).

We hypothesised that because of the slow reaction of protected SOC pools to temperature change, medium-term warmed

soils would still be losing SOC over time, while the long-term warmed soils would have reached a new equilibrium at lower SOC content. We further hypothesised similar subsoil and topsoil SOC loss, given that subsoils were exposed to the same warming intensity and duration as topsoils.

## 2  Large, linear and fast topsoil SOC loss

Topsoil (0-10 cm; comprising the A horizon and rooting zone) SOC stocks linearly declined by $2.8 \pm 0.5$ ($\pm$ SE) ton SOC ha$^{-1}$ °C$^{-1}$

soil warming (or $8.8 \pm 2.1$ °C$^{-1}$; both P < 0.001) for mass-corrected SOC stocks (fig. 1a, 2). These topsoil temperature responses did not differ between the medium-term and the long-term warmed grassland, i.e., the warming:grassland interaction term was not significant (P = 0.47). This clearly suggests that warming induced SOC losses only during the initial five years of exposure and that SOC stocks did not change thereafter.

However, several sources of variation such as sampling errors and the large heterogeneity inherent to soils, induced quite

broad uncertainty intervals that reduced the potential to detect statistically significant changes in SOC stocks or in their temperature response. Hence, to demonstrate that the warming-induced SOC stock loss had indeed stabilised within five years of warming and did not further loose SOC, we calculated what SOC stock decline could have remained undetected given the variability in our samples using a one-sided 95 % confidence interval on the soil warming regression coefficient. This shows that average additional SOC stock losses smaller than 0.88 ton C ha$^{-1}$ °C$^{-1}$ would not have been detected at p<0.05, implying that in the five

year period following the initial warming response (i.e., 2013-2018), annual declines of up to 0.18 ton C ha$^{-1}$ °C$^{-1}$ year$^{-1}$ would have remained undetected. In the subsequent time span of >50 years, only changes smaller than 0.018 ton C ha$^{-1}$ °C$^{-1}$ year$^{-1}$ would have remained undetected, i.e., a rate 30-fold less than the SOC loss observed in the initial 5 years of soil warming.

Also when not corrected for warming-induced density changes, SOC stocks and soil C concentrations declined with warming (fig. B1; fig. B2) and did not further decrease after five years of soil warming. In contrast to our hypothesis, our data thus

revealed that in topsoil, a stepwise increase in temperature caused a fast SOC stock loss that stabilised within five years of





warming, despite the sustained higher temperatures. Even grasslands that had been warmed at least 55 years exhibited no larger SOC loss than that observed after 5 years of soil warming.

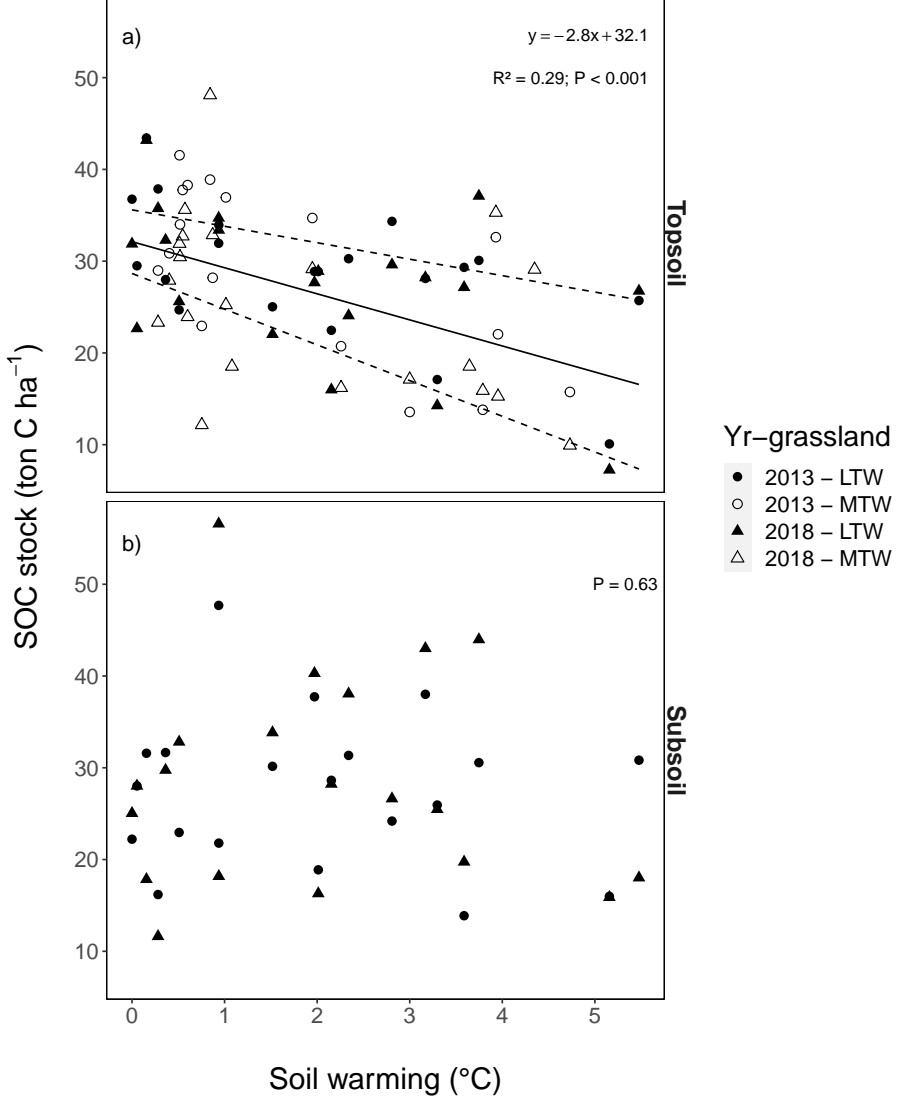

**Figure 1. Soil organic carbon (SOC) stocks (ton C ha$^{-1}$) along soil warming gradients**, a) in the topsoil (0-10 cm) and b) in the subsoil (10-30 cm) after a soil mass correction. The regression (solid line; dashed lines represent the 95 % confidence interval; regression details provided in the inset) for medium-term warmed (MTW) and long-term warmed (LTW) grassland (in topsoil) for both 2013 and 2018 were combined, since no soil temperature x warming-duration interaction effect, nor a main effect for warming-duration or sampling year was found. The soil mass correction is visualised in fig. B8. The uncorrected SOC stocks yield qualitatively similar conclusions (fig. B1), as did the C percentage in top- and subsoil (fig. B2). (n = 78 & 40 for topsoil and subsoil respectively)



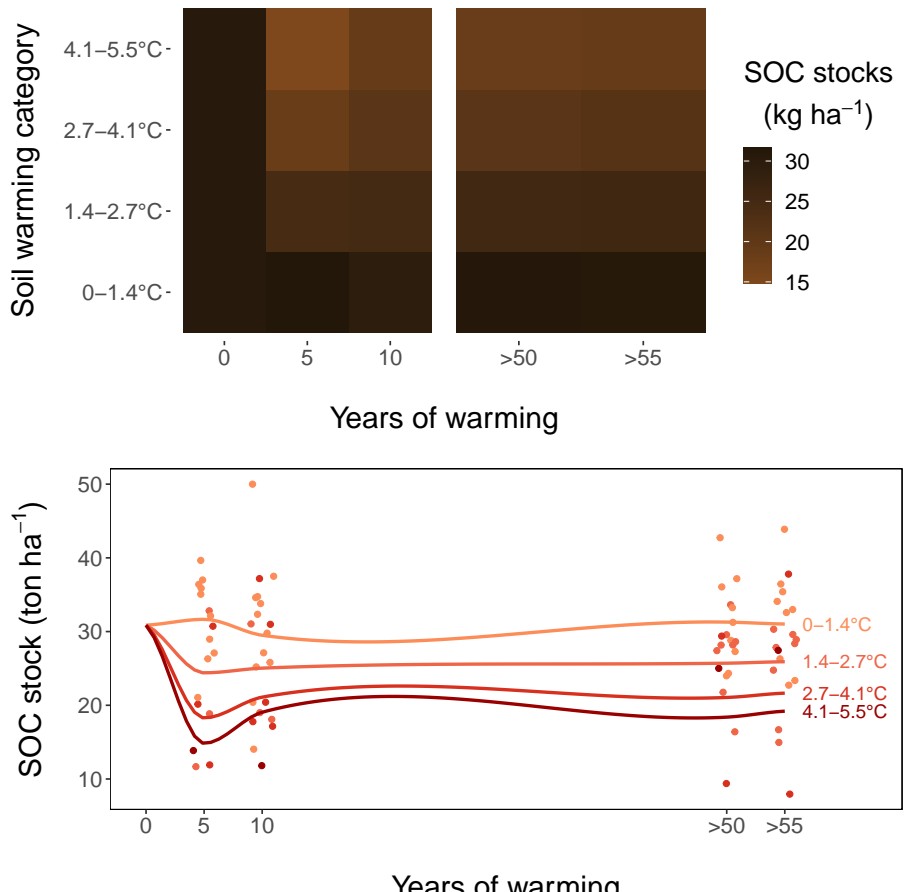

**Figure 2. Warming effect on topsoil (0-10 cm) SOC stocks**, as observed during repeated sampling campaigns. Stocks after 5 and 10 years of warming are sampled in the medium-term warmed grasslands, stocks after >50 and >55 years in the long-term warmed grasslands. The data at the start of warming is interpolated from the ambient plots in grasslands combined. Soils are divided in four warming categories for representation. The colours on the heatmap and the smoother lines are based on a linear regression equation per sampling event.

To gain insight in the warming-induced soil physical changes and their effects on SOC stocks, soils were fractionated into different size classes that were analysed separately (see Methods). Aggregate fractionation showed that with increasing

warming intensity, the mass of >2 mm fraction declined significantly, in favour of the >250 μm and >63 μm fractions. No significant change was detected in the mass of the smallest (<63 μm) fraction (fig. 3). Opposed to this contrasting response of relative mass, all soil fractions exhibited similar soil C % declines with soil warming (fig. 3). The relative mass increase of the smaller fractions was compensated for by the soil C % decline, resulting in a stable amount of C in the >250 μm and >63 μm fractions. As a result, all of the warming-induced SOC stock decline we observed in the bulk soil, was attributable to C losses

in the >2 mm fraction.

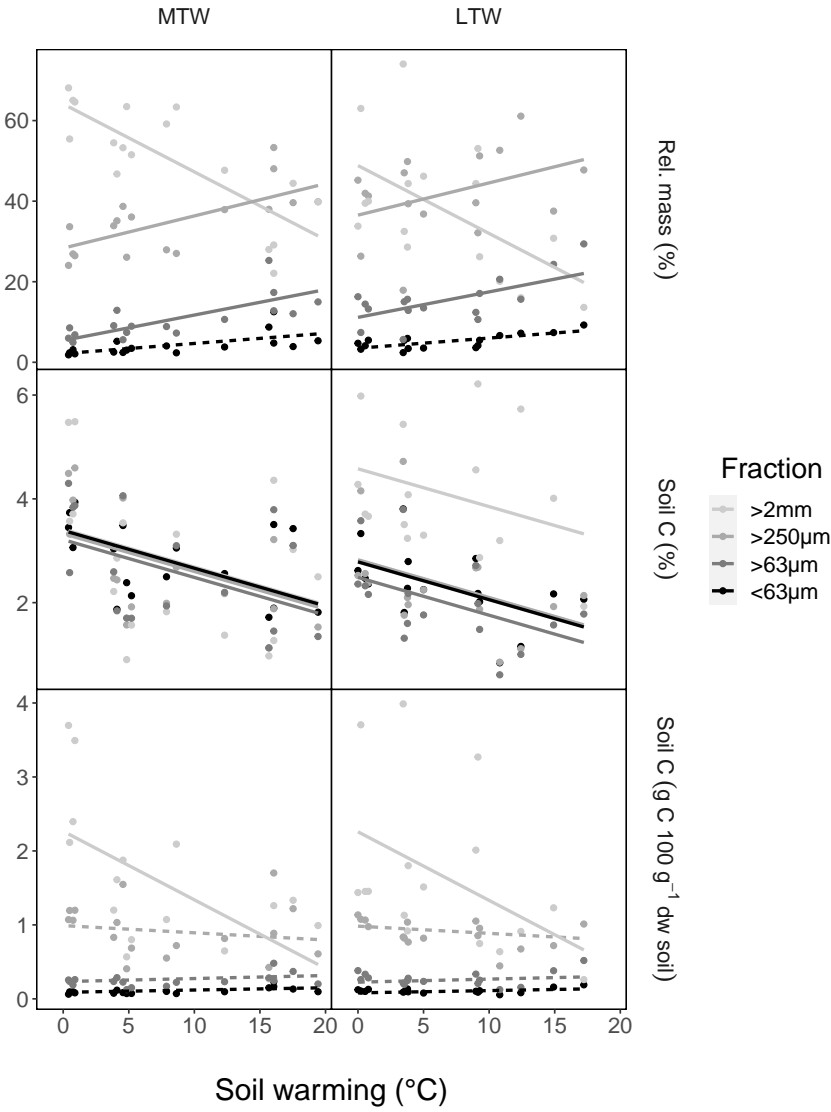

**Figure 3.** Relative mass, soil C % and absolute soil C amount of soil aggregate fractions originating from topsoil in the medium-term warmed (MTW) and long-term warmed (LTW) grassland. Darker lines indicate smaller fractions. Fractions significantly affected by soil warming are represented with solid lines, non-significant relations upon soil warming are shown with dashed lines. (n = 17 & 16 for each fraction in MTW and LTW grassland respectively)

We suggest that the rapid topsoil SOC loss observed under warming, as well as its attenuation in the medium-term, emerged from the interplay between soil microbial biomass and activity. Warming at the same study site accelerated microbial growth and respiration (Marañón-Jiménez et al., 2018; Walker et al., 2020), which, in the absence of increased plant inputs to soil (fig. B3), caused the initial SOC loss observed here (fig. B1a). In turn, the warming-induced SOC loss caused a decline in micro-





bial biomass, creating a negative feedback on microbial activity that we presume prevented further SOC loss (Walker et al., 2018, 2020). Alternatively, ephemeral SOC loss under warming may have resulted from physiological adaptations (Allison et al., 2010; Bradford et al., 2019) or compositional shifts (Melillo et al., 2017) in the microbial community, but we found no evidence this occurred here. Previous research in these grasslands showed that soil microbial carbon use efficiency (CUE) remained constant under short- and long-term warming (Walker et al., 2020), and microbial community composition was only

affected by more intense (>9 °C) long-term warming (Radujković et al., 2018). Aggregate fractionation suggests C % all size fractions were impacted similarly by warming (fig. 3). This likely indicates both particulate organic matter, often occluded in large-size aggregates, and mineral-associated organic matter, present in all aggregate sizes, decreased with warming.

## 3    Stable subsoil SOC stocks

We hypothesised that the similar warming intensity across the soil profile (fig. B4) would elicit similar declines in subsoil

SOC stocks than those in topsoil. In contrast, SOC remained constant in the subsoil, even under 50 years of soil warming in the long-term warmed grasslands (P = 0.63; fig. 1b, 2). This lack of SOC loss from the subsoil may be explained by three, potentially co-occurring mechanisms. First, limited fresh C inputs from litter and root exudates below the rooting zone (<10 cm deep; fig. B3) could be a critical factor (Tian et al., 2016) explaining the lack of a positive warming effect on subsoil decomposition, as a plausible positive priming effect often elicited by fresh C inputs would have been restricted to the topsoil.

Second, a large fraction of SOC in the topsoil is particulate organic matter protected in aggregates, whereas most subsoil SOC is associated with minerals (Fontaine et al., 2007; Rumpel and Kögel-Knabner, 2011). As such, by accelerating mineralisation of plant litter that is deposited only in topsoil (Walker et al., 2018), and thereby reducing aggregate stability and breaking up macroaggregates (fig. 3) (Poeplau et al., 2020), warming may have had much greater effect on SOC losses in topsoil than in subsoil, where mineral protection dominates. Third, although unlikely given the absence of increased dissolved organic C

(DOC) with warming in subsoil (fig. B6), it can also not be excluded that SOC stocks in subsoils only appear stable, because increased losses are compensated for by increased inputs from above (Osher et al., 2003).

Only very few studies have assessed subsoil SOC stocks responses to deep soil warming, and observed responses differ strongly in magnitude and even direction. Higher subsoil than topsoil SOC losses were reported in two forests (Lin et al., 2018; Soong et al., 2020a), while unresponsive (this study) subsoil SOC stocks or even increases in subsoil SOC stocks were observed

in grasslands (Jia et al., 2019). Further research is needed to unravel the drivers of these contrasting subsoil SOC responses to warming among experiments, which may be related to differences in soil properties, aggregate dynamics or rooting depths.

## 4    Implications for carbon-climate feedbacks

Earth System Model (ESM) inter-comparison studies (Eyring et al., 2016) have revealed large variability in both contemporary global SOC stock estimates and future SOC stock projections, underlining the need for empirical observations to better con-

strain the response of SOC to temperature change (Nishina et al., 2014). Long-term warming experiments like this study are





thus needed to reduce the uncertainty on model projections (Abramoff et al., 2019). Although geothermally active areas offer long-lasting, continuous and large soil temperature gradients and overcome the technical challenges and high costs associated with warming manipulation experiments (Sigurdsson et al., 2016; O'Gorman et al., 2014), their use as a proxy for climate change has some drawbacks of its own, such as limited aboveground warming and a stepwise increase in soil temperature at

the initiation of the geothermal gradient (De Boeck et al., 2015). Also the Andosol, covering only $\pm$ 0.8 % of the earth's surface (Baillie, 2001), makes that one should be cautious extrapolating the results to the entire sub-arctic region. Nonetheless, this site offers a unique opportunity to study the direct versus long-term response of SOC stocks to temperature change and the results from this study and other deep soil warming experiments clearly indicate that introducing vertically resolved plant- and microbial dynamics in ESMs is a necessity for more accurate projections of the carbon-climate feedback.

In conclusion, warming caused a large but rapidly equilibrating SOC loss in the topsoil that increased linearly with warming intensity, while no SOC loss was observed in the subsoil in our subarctic grasslands exposed to decades of soil warming. Future work should focus on understanding whether these observed temporal dynamics are consistent throughout the northern non-permafrost region. Improved understanding of the variation in subsoil SOC responses to warming is also critical for constraining Earth System Models and obtaining reliable climate projections.





**Appendix A: Material and methods**

This study was conducted at the ForHot research site, located in the Hengill geothermal area, 40 km east of Reykjavík, Iceland
(64°00'01" N, 21°11'09" W; 100 – 225 m a.s.l. (Sigurdsson et al., 2016). The mean annual temperature between 2006 and 2016
was 5.2 ± 0.1 (± SE) °C, and mean annual daily minimum and maximum temperatures were 2.2 ± 0.2 (SE) and 8.6 ± 0.2 (±
SE) °C. The mean annual precipitation during the same period was 1413 ± 57 (± SE) mm (Icelandic Meteorological Office;
Eyrarbakki weather station which closed in 2017). The main vegetation type is unmanaged grassland, dominated by *Agrostris
capillaris*, *Ranunculus acris* and *Equisetum pratense* and the underlying soil is classified as Brown Andosol (Arnalds, 2015).

In this study, we define the 0-10 cm layer as topsoil, and the 10-30 cm layers as subsoil. This subsoil layer strongly differed
from the topsoil in many ways. First in these soils, the A horizon that is enriched with SOC is maximum 10 cm deep (Arnalds,
2015). Second, 95.7 ± 0.4 (SE) % of the fine root biomass sampled in the top 30 cm layer, was found in upper 10 cm (fig. B5).
Third, bulk density in subsoil is significantly higher than in topsoil (P < 0.001) (fig. B7).

The site comprises two areas that have been subjected to geothermal soil warming for different periods of time (Sigurdsson
et al., 2016). One area (hereafter "medium-term warmed grassland") has been warmed since May 2008, when a large earthquake
shifted geothermal systems to previously unwarmed soils. The second area (2.5 km North-east from the first area; hereafter
"long-term warmed grassland") was already mentioned to be warmed in the early 18th century (Magnússon and Vídalín, 1708)
and has thus likely been warmed for centuries. For sure, the warming was registered in a census during the 1960s, and no change
in the location of the hotspots has been recorded during the past 50 years (Kristján Sæmundsson, personal communication).
The soil warming increment at both sites is relatively constant throughout the year and extreme deviations are rare (Sigurdsson
et al., 2016). Soil warming is caused by horizontal heat conduction through the soil, causing fairly homogeneous warming with
depth and inducing a fairly natural temperature depth profile (fig. B4). This homogeneous soil warming is in line with CMIP5
predictions of rapid transfer of the temperature signal from air to shallow and deeper soils (Soong et al., 2020b). The geothermal
water is confined within the bedrock and no signs of soil contamination by geothermal byproducts have been found (Shi et al.,
2020). Soil pH (mean: 5.5 ± 0.1 (SE)) and soil moisture did not show major changes along the soil warming gradients, with
soil moisture very rarely approaching the permanent wilting point and no relation between soil temperature and the frequency
of drought events (Leblans et al., 2017). Further, the plant species composition was very similar between the medium-term and
the long term warmed grassland and no drastic changes in dominant plant species occurred up to +6.4 °C warming (Leblans
et al., 2017) (which is the RCP8.5 projected annual warming level for high northern latitudes for the year 2100) (IPCC, 2013).
More detailed information on the site characteristics can be found in Sigurdsson et al. (2016).

We established five replicate transects in each area (the medium-term and the long-term warmed grassland) in 2012, around
two and four geothermal heat sources respectively. In the medium-term warmed grassland, all transects were located on south-
west facing slopes, three with the geothermal heat source at the bottom of the slope, and two with the geothermal heat source
at the top, to eliminate effects of topography and downward transport of groundwater and nutrients from introducing a bias
in the SOC stocks. In the long-term warmed grassland, all transects were located on level ground. Within the long-term and
medium-term warmed grassland, all measurement plots had similar microtopography, soil depth and grazing history.



Each transect consists of six 2 x 2 m permanent measurement plots distributed along the soil temperature gradient, including
unwarmed soil (MAT: 5.7 ± 0.1 °C), yielding 60 plots in total. Each 2 x 2 m permanent measurement plot was accompanied by
two adjacent 0.5 x 0.5 m subplots for destructive measurements. Plot-specific soil warming was recorded hourly at 10 cm soil
depth using HOBO TidbiT v2 Water Temperature Data Loggers (Onset Computer Corporation, USA). Because the permanent
plots occurred at different warming intensities in the different transects, we adopted a regression approach (see statistics below).
More detailed information on the experimental design is provided in Sigurdsson et al. (2016).

In July 2013 and 2018, two 0-10 cm soil cores (corer ø= 5.12 cm) were taken within each subplot. In the medium-term
warmed grassland, soils were too shallow to sample deeper, but additional 10-30 cm cores were taken in the long-term warmed
grassland. Cores were analysed for: (1) soil C concentrations; (2) pH (topsoil only); (3) soil bulk density (BD); and (4) SOC
stocks.

From the first core we obtained fine roots (<2 mm) and soil particles (>2 mm) (necessary to calculate BD) by washing the
cores over two sieves with mesh sizes 2 mm and 0.5 mm. Roots and >2 mm particles were dried and weighed to gain fine root
biomass (g m$^{-2}$) and the volume of >2 mm particles (g cm$^{-3}$) was measured by the water displacement method. The second
soil core was first dried and weighed (as for aboveground vegetation), and soil was then sieved to obtain soil particles <2 mm
and split into three aliquots. One aliquot of 2 g was milled (Retsch MM301 Mixer Mill, Haan, Germany) and analysed for
C concentration (%) by dry combustion (Macro Elemental Analyser, model vario MAX CN, Hanau, Germany). Finally, BD
(g cm$^{-3}$) and SOC stocks (ton ha$^{-1}$) were calculated according to the approach described in Bárcena et al. (2014).

To measure dissolved organic matter (DOC), teflon suction cup lysimeters (Prenart Super Quartz, Prenart Equipment Aps,
Frederiksberg, Denmark) were placed at about 30-40 cm depth in the medium-term and long-term warmed grassland, in Octo-
ber 2014. Samples were taken during summer 2015, 2016 and 2017. The DOC was analysed with a combined Total Organic
Carbon (Shimadzu, Kyoto, Japan). Further installation details of the lysimeters are described in Edlinger (2016), as well as
the sampling procedure. The C-input data for arbuscular mycorrhizae originates from Zhang et al. (2020), where the sampling
procedure is described. Aboveground biomass was sampled by placing a 20x40 cm frame on the plot, after which all vegetation
was clipped. The samples were taken to the lab and sorted by hand in a grass and moss fraction. Both fractions were dried for
48 h at 70 °C, weighed and milled. The samples were analysed for C concentration (%) by dry combustion (Macro Elemental
Analyser, model vario MAX CN, Hanau, Germany). Aggregate fractionation was done in 2018 only. Per plot, a 0-10 cm soil
core was taken (corer ø= 5.12 cm) and dried at room temperature for a some weeks. Stones were removed and aggregates
larger than 8 mm were broken up by dry-sieving on a 8 mm soil sieve. The dry-sieved soil was then slaked for 5 min with DI
water, after which it was wet-sieved on a 2 mm, 250 µm and 63 µm, to separate into four size fractions. Each fraction was dried
at 70 °C for 72 h, after which all fractions were ground with a ball mill to homogenise and analysed for C concentration (%)
by dry combustion (Macro Elemental Analyser, model vario MAX CN, Hanau, Germany). Relative mass of the fractions was
calculated by dividing the fraction mass by the sum of all fraction masses of initial sample. The absolute soil C-amount of each
fraction was calculated by multiplying the fraction soil mass per 100 g of dry soil with the soil C % (fig. 3).

A soil mass correction of the SOC stocks as described in Ellert and Bettany (1995) was necessary to compare stock changes
across the soil warming gradient as soil compaction increased with warming in the upper soil layers (increasing BD; fig. B7),





implying that soil depths in unwarmed soil corresponded to shallower soil depths at warmer soils. The calculation method is
explained in detail in fig. B8.

The soil warming dependence of bulk soil SOC stocks (corrected and uncorrected for soil compaction), BD, DOC and soil C
% were tested with a linear mixed effects model (Pinheiro et al., 2021). Soil warming and warming-duration (medium-term vs.
long-term) were included as main effects, while sampling year was used as random effect to account for sampling differences
and interannual variabilities between the two sampling campaigns. In all cases, criteria for normality and homoscedasticity
were met. For all tests, the dataset was reduced to cover only the warming levels captured by the projections for high northern
latitudes for the year 2100 (0 – 6.4 °C warming) (IPCC, 2013). All tests were performed using R software (R Development
Core Team, 2011).





## Appendix B: Supplementary

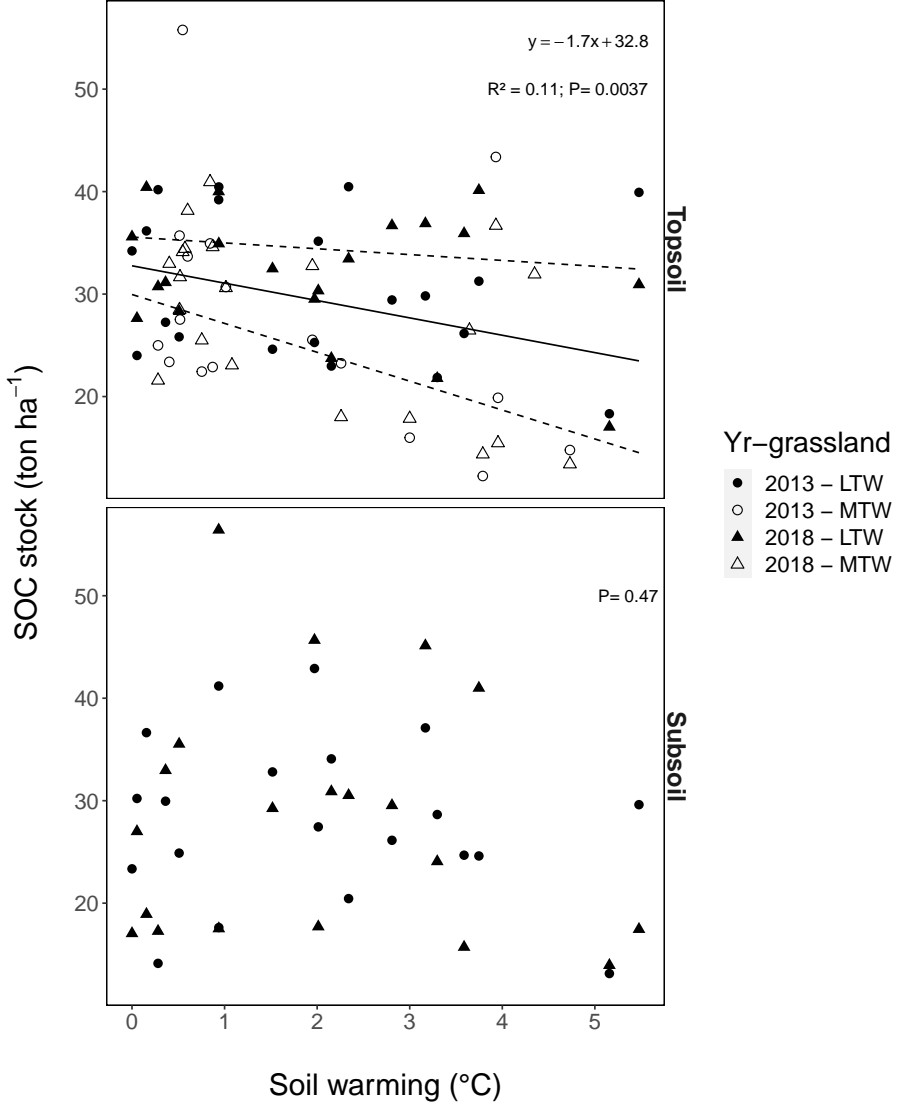

**Figure B1.** Reduction of soil organic carbon (SOC) stock with soil warming, a) in the topsoil (0-10 cm); b) in the subsoil (10-30 cm). All soil samples were taken in July 2013 or July 2018. The regression for medium-term warmed (MTW) and long-term warmed (LTW) grassland (in topsoil) for both 2013 and 2018 were combined, since no soil temperature x warming-duration interaction effect, nor a main effect for warming-duration was found. Soil warming is expressed relative to ambient soil temperature (both at 10 cm depth). In topsoil, a linear relation was observed, while no significant effect was present in subsoil. The 95 % confidence bounds are shown around the topsoil regression slope. (n = 78 & 40 for topsoil and subsoil respectively)



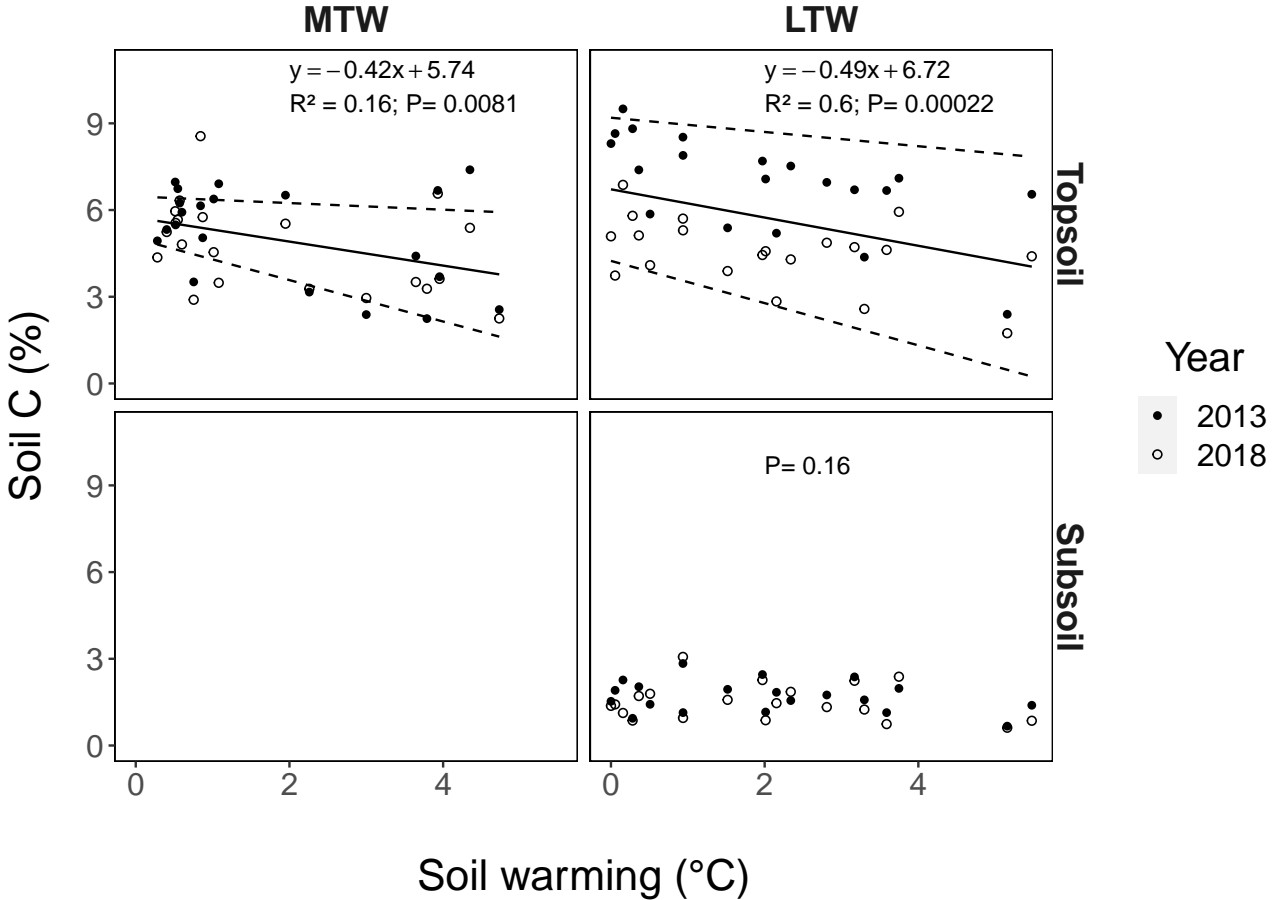

**Figure B2.** Percentage of carbon in topsoil and subsoil, for the long-term warmed (LTW) and medium-term warmed (MTW) grassland. The 95 % confidence bounds are shown around the topsoil linear regression slopes. (topsoil n = 42 & 40 for MTW and LTW grassland respectively; subsoil n = 40 for LTW grassland)





**Figure B3.** Proxies for annual soil C-inputs from aboveground biomass (AGB), fine root biomass and arbuscular mycorrhizae (AMF) in the medium-term warmed (MTW) and the long-term warmed (LTW) grasslands. Vascular plant aboveground biomass was used as a proxy for aboveground C-inputs; vascular plant fine root biomass for belowground C-inputs and C sequestered by AMF (Cnew; data from Zhang et al. (2020)) for C-inputs by arbuscular mycorrhizae. For the observed soil warming range (0-6.4 °C warming), no change in C-inputs could be found. P-values were obtained by a linear regression analysis. (n = 20 & 17 for MTW and LTW inputs respectively)





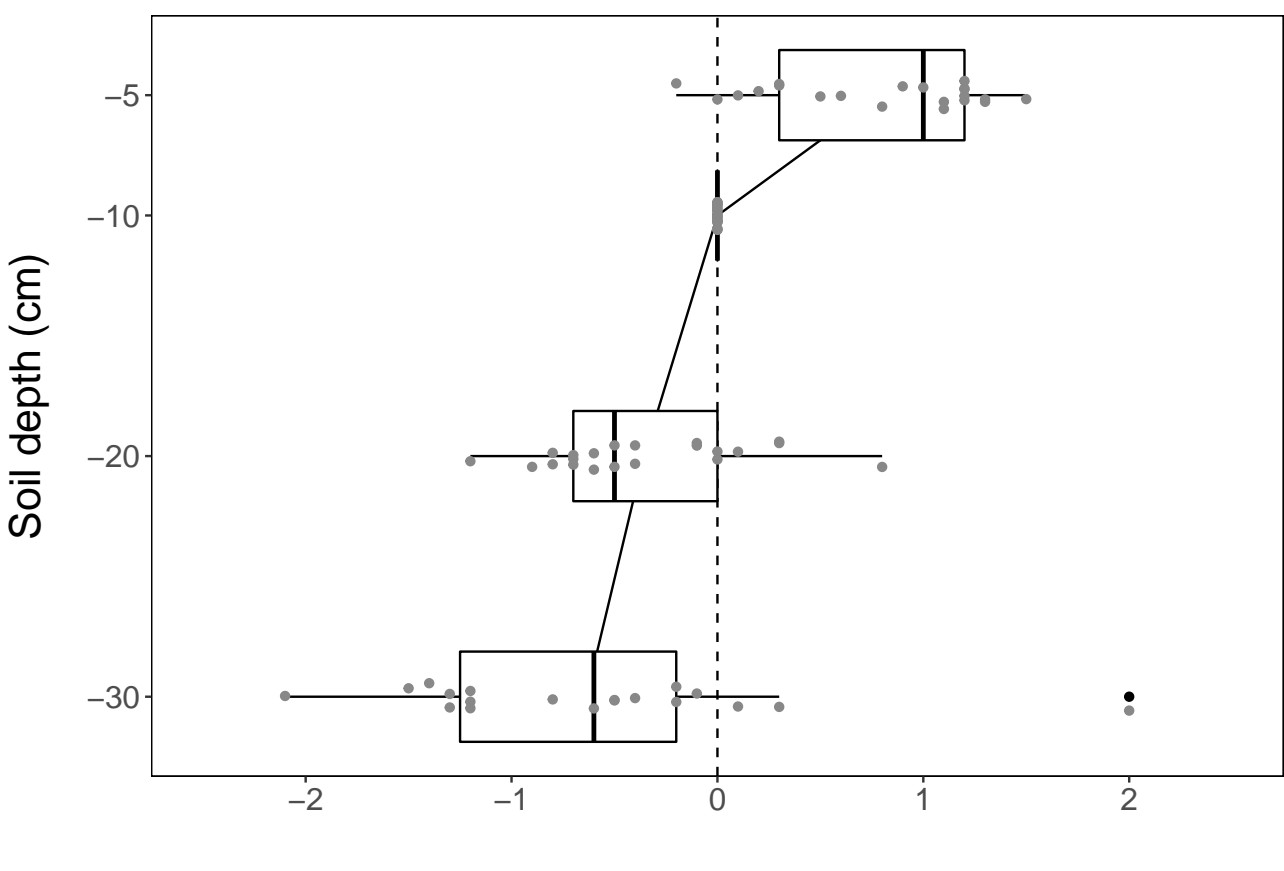

**Figure B4.** Soil warming along the vertical soil profile, based on the reference temperature measured at 10 cm which is used throughout the paper. The median temperature is about 1 °C higher at 5 cm depth, while being slightly lower at 20 cm and 30 cm depth. Due to the shallow soil in the medium-term warmed grassland, the warming profile is given for the long-term warmed grassland only.



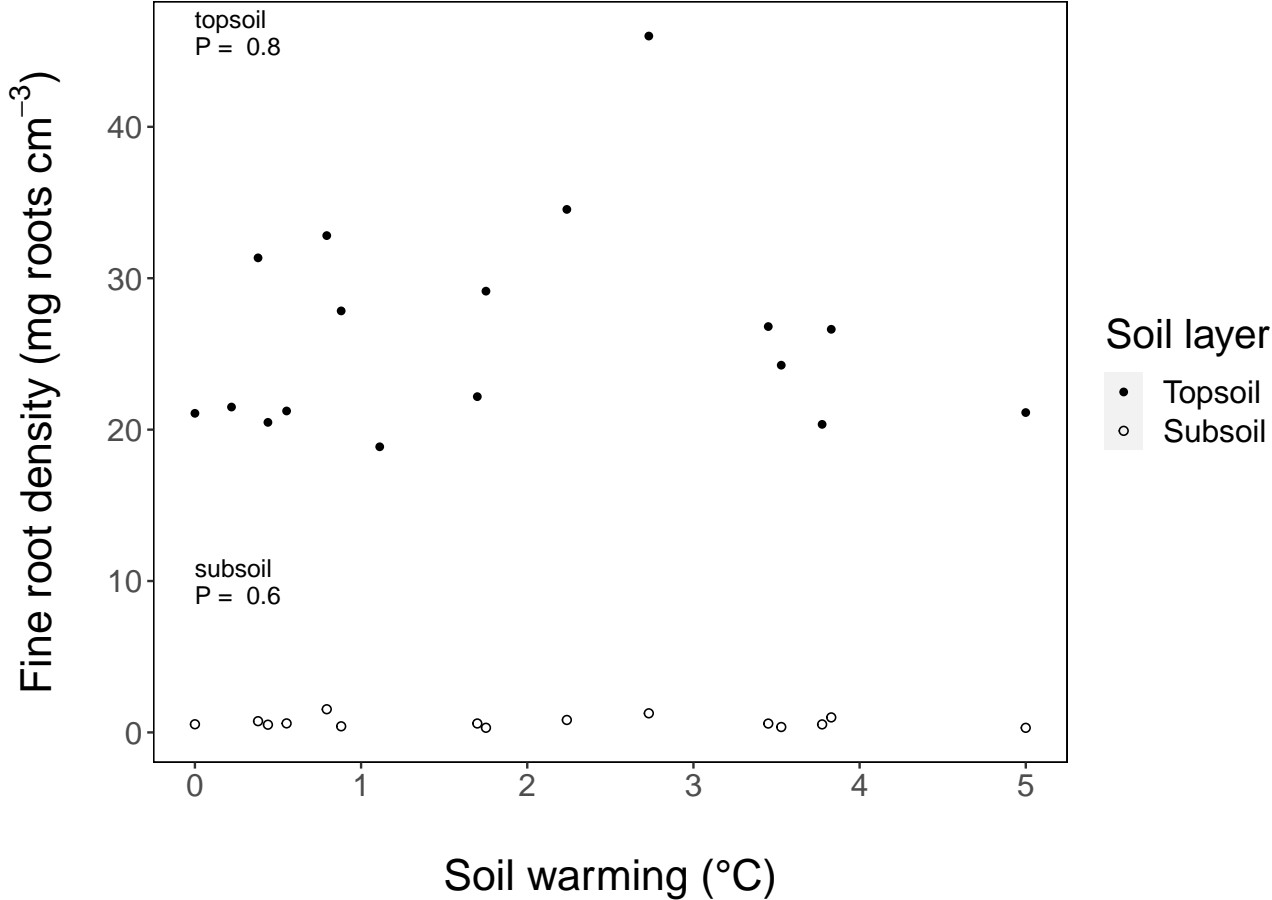

**Figure B5.** Fine root density in long-term warmed (LTW) grassland in top-and subsoil sampled in July 2018. For the observed soil warming range (0-6.4 °C warming), no change in C-inputs could be found. P-values were obtained by a linear regression analysis. In the top 30 cm soil layer, 95.7 ± 0.4 (SE) % of the fine root biomass was found in upper 10 cm, which is why we define the 0-10 cm layer as topsoil, and the 10-30 cm layer as subsoil. (n = 17 & 15 for topsoil and subsoil respectively)



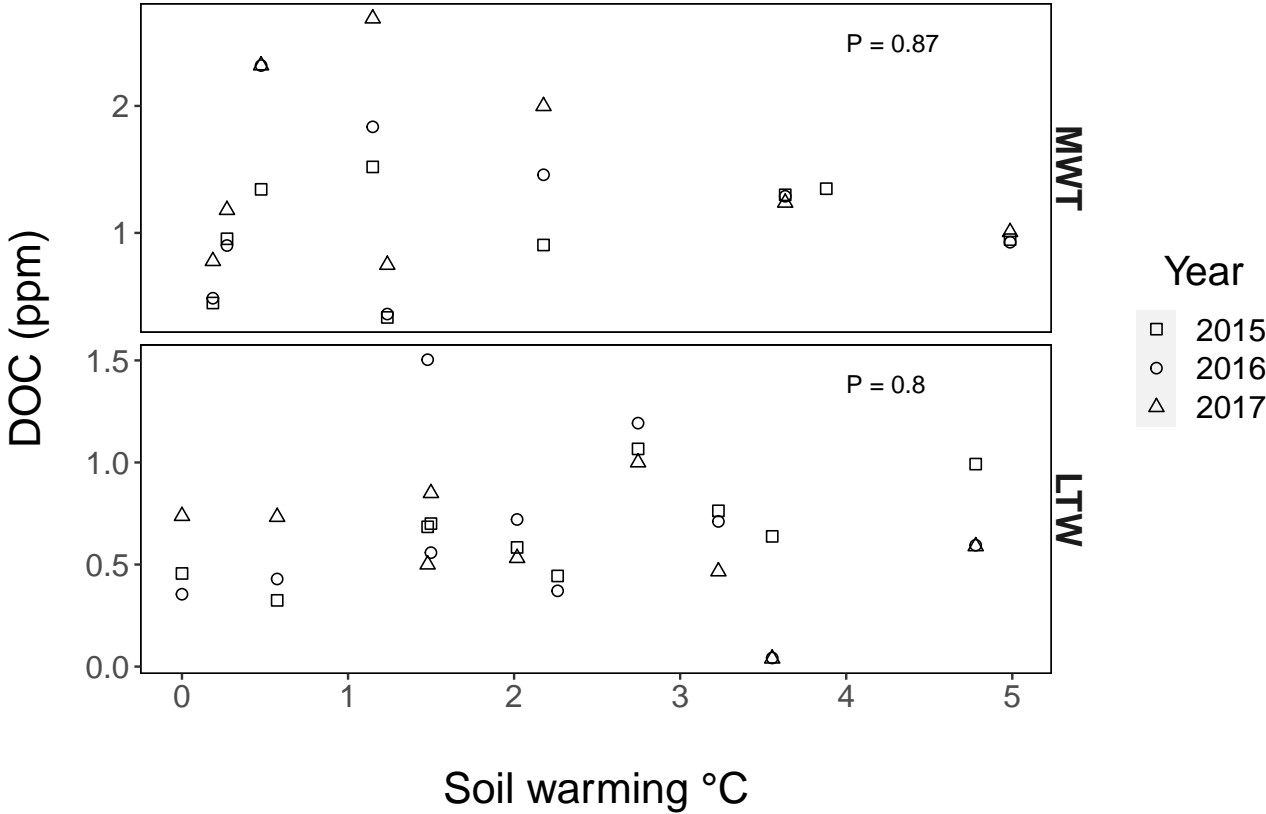

**Figure B6.** Dissolved organic carbon (DOC) in medium-term warmed (MTW) and long-term warmed (LTW) grassland during summer 2015, 2016 and 2017. No DOC change was observed with soil warming in the SWT (P = 0.95), nor in the LTW (P = 0.8), meaning that carbon-inputs into deeper soil layers remained constant over the whole soil temperature gradient. Statistical analysis was done with a linear mixed effects model with soil warming as explanatory variable and sampling year as a random factor. Criteria for normality and homoscedasticity were met. Dissolved organic carbon was sampled at a approximate depth of 30 cm with Prenart Super Quartz (Prenart, Fredriksberg, Denmark) soil water samplers, installed around 1 October 2014. The samples was always inserted from 'downslope,' where the soil was deep enough. Analyses of the samples was done at the IGN Biochemistry Lab, University of Copenhagen. (n = 25 & 29 for MTW and LTW grassland respectively)



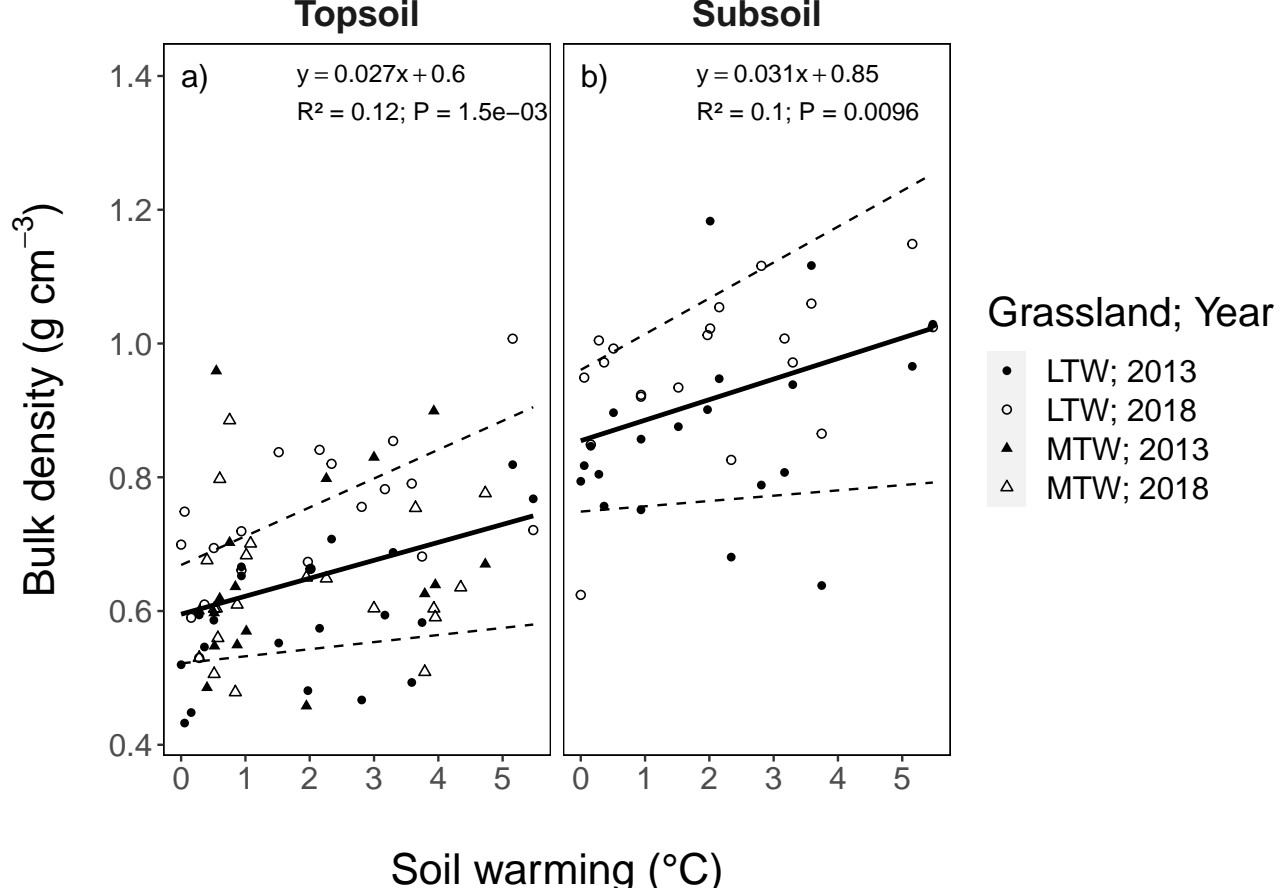

**Figure B7.** Changes in bulk density with soil warming in the long-term warmed (LTW; circles) and the medium-term warmed (MTW; triangles) grassland. Soil warming is expressed relative to ambient soil temperature (at 10 cm depth). Bulk density is separated for the topsoil (0-10 cm) and the subsoil (10-30 cm). The regression for medium-term warmed (MTW) and long-term warmed (LTW) grassland (in topsoil) for both 2013 and 2018 were combined, since no soil temperature x warming-duration interaction effect, nor a main effect for warming-duration or sampling year was found. The 95 % confidence bounds are shown around the regression slopes. (n = 78 & 40 for topsoil and subsoil respectively)



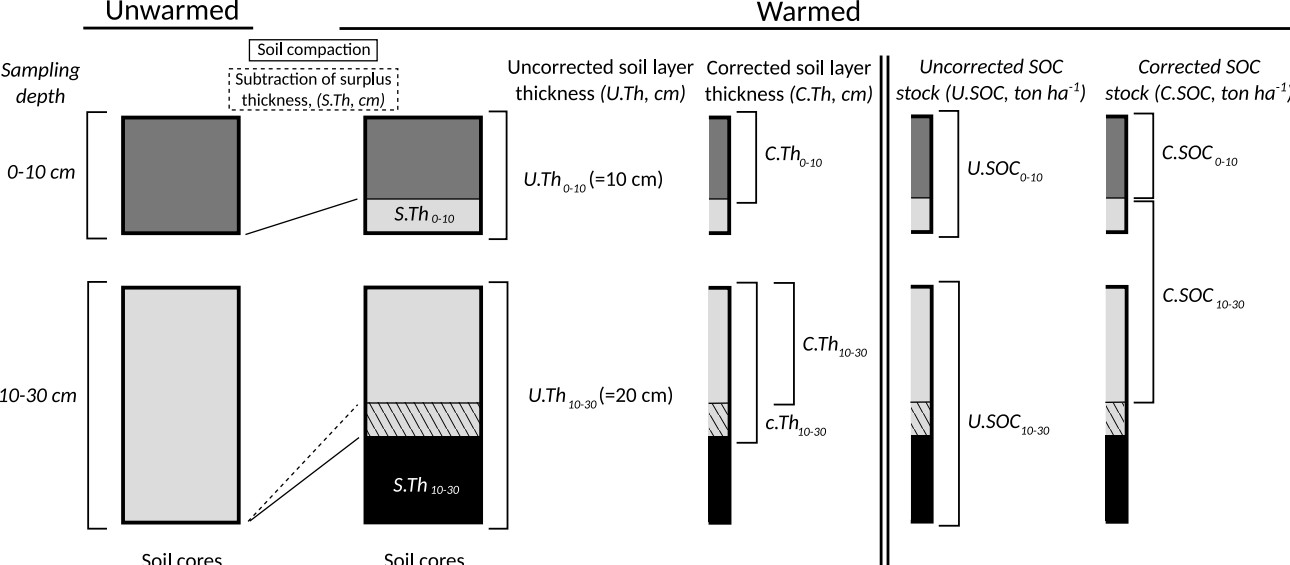

**Figure B8. Soil mass correction.** Due to significant soil compaction (increasing bulk density (BD)) with increasing soil temperature in the upper soil layers (fig. B7), a certain soil depth in unwarmed soil corresponds to ever shallower soil depths at warmer soil temperatures. Therefore, the SOC stocks were corrected for soil compaction, i.e., the corrected SOC stocks were calculated on the same mass of soil.

For topsoil, we calculated a corrected thickness of a warmed soil layer, corresponding to the same core mass as the the unwarmed soils. Using the ratio of corrected and uncorrected layer thickness, we calculated a corrected SOC stock for the warmed topsoil. For warmed subsoils, we calculated the corrected thickness in the same way as for topsoil, but subtracted a surplus thickness of the above topsoil. The corrected subsoil SOC stock was then calculated as the sum of the surplus topsoil SOC stock and the SOC stock in the corrected subsoil layer. The detailed calculation method is shown below.

**Corrections 0-10cm soil layer**

First, the corrected soil layer thickness of the 0-10 cm layer ($C.Th_{0-10}$) was calculated for the warmed soils:

$$C.Th_{0-10} = \frac{U.Th_{0-10} \times U.BD_{0-10}}{C.BD_{0-10}} \tag{B1}$$

Where $U.Th_{0-10}$ is the uncorrected soil layer thickness of the 0-10 cm layer (10 cm), $U.BD_{0-10}$ is the uncorrected BD of the 0-10 cm layer (which corresponds to the BD at ambient soil temperature) and $C.BD_{0-10}$ is the measured BD for the 0-10 cm layer. Then, the corrected SOC stocks of the 0-10 depth layer ($C.SOC_{0-10}$) were calculated:

$$C.SOC_{0-10} = \frac{U.SOC_{0-10} \times C.Th_{0-10}}{U.Th_{0-10}} \tag{B2}$$





Where $U.SOC_{0-10}$ is the uncorrected SOC stock in the 0-10 cm depth layer, and $C.Th_{0-10}/U.Th_{0-10}$ corresponds to the proportional thickness of the corrected layer compared to the uncorrected layer.

**Corrections 10-30 cm soil layer (only applicable to the long-term warmed grassland)**

First, the thickness of the surplus soil layer from the 0-10 cm layer ($S.Th_{0-10}$) was calculated:

$$S.Th_{0-10} = \frac{(U.Th_{0-10} - C.Th_{0-10}) \times U.BD_{0-10}}{C.BD_{0-10}} \tag{B3}$$

The second term is a correction factor for the soil compaction of the surplus soil layer. Then, the corrected thickness of the 10-30 cm soil layer, not yet taking the surplus soil sampled from the 0-10 cm layer ($S.Th_{0-10}$) into account, ($c.Th_{10-30}$) was calculated:

$$c.Th_{10-30} = \frac{U.Th_{10-30} \times U.BD_{10-30}}{C.BD_{10-30}} \tag{B4}$$

Where $U.Th_{10-30}$ is the uncorrected soil layer thickness of the 10-30 cm layer (20 cm), $U.BD_{10-30}$ is the uncorrected BD of the 10-30 cm layer (which corresponds to the BD at ambient soil temperature) and $C.BD_{10-30}$ is the measured BD for the 10-30 cm layer.

Subsequently, we took into account the thickness of the surplus soil sampled from the 0-10 cm layer to calculate the final corrected soil thickness of the 10-30 cm soil layer ($C.Th_{10-30}$). Hence, $C.Th_{10-30}$ is the part of the 10-30 cm layer that remains after (i) correcting for soil compaction and (ii) subtracting the thickness of the surplus soil sampled at the 0-10 cm layer:

$$C.Th_{10-30} = c.Th_{10-30} - S.Th_{0-10} \tag{B5}$$

Subsequently, the corrected SOC stock for the 10-30 cm layer ($C.SOC_{10-30}$) was calculated:

$$C.SOC_{10-30} = (U.SOC_{0-10} - C.SOC_{0-10}) + \frac{U.SOC_{10-30} \times C.Th_{10-30}}{U.Th_{10-30}} \tag{B6}$$

Where $U.SOC_{10-30}$ is the uncorrected SOC stock in the 10-30 cm depth layer, and $C.Th_{10-30}/U.Th_{10-30}$ corresponds to the proportional thickness of the corrected layer compared to the uncorrected layer.

*Data availability.* All data used in this manuscript has been made available on Zenodo. DOI: 10.5281/zenodo.4745479

*Author contributions.* BDS, NIWL, SV, JLS, LF, JTW, HW, PMVB, NV and IAJ designed the study. PG, HW, CF, NIWL, BDS, IAJ, SV, KVDV, EV, ZFL, SM-J, NV and MM provided the data. All authors contributed substantially to the analysis and the writing of the manuscript.





*Competing interests.*   The authors declare that they have no conflict of interest.

*Acknowledgements.*   This research was supported by a joint Fonds Wetenschappelijk Onderzoek Flanders (FWO) and Fonds zur Förderung der wissenschaftlichen Forschung (FWF) grant with nos. FWO-G0F2217N & FWF-I-3237, awarded to I.A.J and M.B., the European Research Council Synergy grant 610028 (IMBALANCE-P), and the Research Council of the University of Antwerp (FORHOT TOP-BOF project). This work contributes to the FSC-Sink, CAR-ES and the ClimMani COST Action (ES1308). Reykir - the Icelandic state gardening

school, Keldnaholt – the Agricultural University of Iceland and Mogilsá – the Icelandic Forest Research, provided logistical support for the present study. Further, we thank the Lorentz Center in Leiden. We thank Iris Janssens, Jochen Janssens, Inge Van De Putte, Maxime Sepelie, Jana Vynckier, Alexander Meire, Lieven Michielsen, Freja Dreesen, Sebastien Leys, Elín Gudmundsdóttir, Annemie Vinck, Paul Leblans, Kwinten Leblans, Sigvatur, Már Gudmundsson, Elías Óskarsson and Simon Arnar Pálsson for their help in the field. We thank Brita Berglund, Baldur Vigfusson, Nadine Calluy, Tom Van Der Spiet, Anne Cools, Marijke Van den Bruel, Els Oosterbos, Saad El-Rawi, Miguel

Portillo Estrada and Wannes Kiebooms for their assistance with the lab analyses.



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
