# Peer review of "Soil carbon loss in warmed subarctic grasslands is rapid and restricted to topsoil"

_Biogeosciences, 2021_

## Author Response (AR1)

**RC1: 'Comment on bg-2021-338', Emma Sayer, 01 Feb 2022 reply**

**The manuscript presents a study of SOC stocks to 30-cm depth in a unique long-term soil warming study based on natural geothermal gradients. The manuscript presents differences in SOC stocks along warming gradients at sites that had been warmed for 5-10 years (medium-term) or 5-55 years (long-term) at the time of the study. The key finding is that C stocks in the topsoil declined with increasing temperature after 5-10 years of warming, but not after 50-55 years, and subsoil C stocks (long-term site only) did not differ with temperature.**

**I read the manuscript with great interest, as data on changes in subsoil C stocks are rare, as are such long-term climate change experiments. The topic is certainly appropriate for Biogeosciences and although the overarching research questions and hypotheses are interesting, I am not entirely convinced that the present study address them in full.**

*We thank the reviewer for the thorough reading of our manuscript and its critical evaluation. We have tried to answer the questions in the best possible way and edit the manuscript accordingly.*

**First, the chosen soil sampling depth needs to be justified, as many researchers would not consider 10-30 cm depth as 'subsoil'.**

*We agree that 10-30 cm is often not considered as 'subsoils', but in our study there were good arguments to do so. The justification of the chosen soil sampling depth for topsoil and subsoil is given in the material and methods section on lines 132-135. Differentiation between topsoil and subsoil is based on three strong differences between the two layers: lower soil C% (figure B2), lower fine root density (figure B5) and higher bulk density in subsoil (figure B7).*

*To increase clarity about how we differentiated between topsoil and subsoil, we added a reference to the justification in the introduction of the main text on line 34-35: "An elaboration on the choice of these two soil layer depths is provided in the material and methods section in appendix A."*

**Second the 10-30 cm depth was only sampled at the site with long-term warming. So it is entirely unknown whether the deeper soils experienced similar short-term effects to the topsoil.**

*We acknowledge the shortcoming of our experimental design, but the shallow soil depth in the medium-term warmed grassland did not allow us to sample below the topsoil. In our manuscript we therefore only report about the long-term warming effects on subsoil. We believe the finding about stable subsoil subject to long-term warming is relevant also without knowing the short- or medium-term warming effects.*

**In addition, there are a few key points that should be considered in the interests of clarity and scientific rigour:**

**First, the language of the present manuscript is somewhat misleading because 'SOC losses' are referred to throughout, but actual C loss was not measured. The space-for-time approach used in the study demonstrates differences in SOC stocks between plots that have been warmed to varying degrees for different lengths of time, which is not the same as measuring C losses. The language in the text should be edited to reflect this.**

*The reviewer is right that we did not measure the actual C losses, which occur through both leaching and respiration. For estimation of the C loss through respiration, eddy covariance would be needed, which is impossible to apply on these small scale warming gradients.*

*To make sure the used terminology corresponds with the actual processes measured, we changed 'SOC losses' to 'SOC stock reductions' throughout the manuscript.*

**Second, the reasoning behind the hypothesis of long-term warming needs a justification – is 50 years sufficient for a new equilibrium to be reached? And why would you expect the new equilibrium to be reached at lower SOC content? Theoretically, maintenance of SOC stocks might also be predicted over the longer term if, e.g. acclimation of microbial communities and C turnover rates, increased plant productivity, or declining nutrient availability with long-term warming eventually compensate for initial losses... A rationale for this hypothesis could be provided by drawing on previous research from the site (currently discussed on lines 76-85).**

*We fully agree that the processes raised by the referee could lead to a maintenance of the SOC content, but at the same time many studies have suggested a larger temperature sensitivity of SOC decomposition than of primary productivity. We added a sentence on this to introduce the hypothesis on line 17-18: "However, soil warming could also be expected to result in increased and/or unaltered SOC stocks, if, e.g., there is rapid acclimation of microbial communities or if plant productivity increases strongly."*

*Additionally, we assume 50 years will suffice for the ecosystem to reach a new equilibrium. If not, the system should at least be very close to equilibrium. However, because any statement on this would be pure speculation, we did not change add text to discuss whether or not 50 years suffice.*

**Third, the introduction states clearly that the processes involved in SOC formation and mineralisation are rarely studied below 20-30 cm depth, which sounds like a justification to study subsoils below 30-cm depth - so why was only 0-30 cm considered in this study? The justification for the split between 0-10 and 10-30 cm is given, but it does not explain why the 10-30 cm increment should be considered as subsoil, nor whether it is likely to be representative of subsoil at greater depths. Indeed, subsoils are often considered as being below 30-cm depth.**

*Indeed, subsoils are often regarded being below 30 cm depth. For the three reasons given before however, we believe the 10-30 cm depth can be considered as subsoil at our experimental site. We further agree with the reviewer that studying soil layers deeper than 30 cm would also be extremely interesting. A new PhD student is currently studying deep soil SOM stocks, and the preliminary measurements in the LTW grassland did not reveal changes in SOC at greater depths.*

**Finally, I'm not entirely convinced by the argument that C inputs did not increase with warming. If I understand correctly, the evidence for the lack of changes in C inputs (presented in figure B3 and B5) is based on measurements and samples collected after the first 5 years of warming. So how do you know there were no short-term changes in C inputs during the first 5 years? The major losses in the first 5 years could be an artefact of the sudden increase in soil temperature – what precludes a similar short-term increase in plant inputs? At the long-term site, I also wondered whether gradual change in temperature and growing season length over the last 50 years could have partially compensated for early C losses?**

*From the start of the experiment, we continuously measured LAI and NDVI, and observed no increases in the seasonal maximum. Hence, we can preclude a short term increase in plant productivity and soil C inputs.*

    *As the reviewer indicates, a short-term overshoot in many ecosystem parameters and a (partial) mitigation in response to long-term warming was documented in Walker et al. (2020). However, when focusing on SOC stock loss in that paper, there was no observable, statistically significant SOC increase after five years of warming. Hence, we did not further elaborate on a possible mitigation after five years of warming, but rather focussed on a rapid SOC stock loss in topsoil.*

*Walker, T. W., Janssens, I. A., Weedon, J. T., Sigurdsson, B. D., Richter, A., Peñuelas, J., ... & Verbruggen, E. (2020). A systemic overreaction to years versus decades of warming in a subarctic grassland ecosystem. Nature ecology & evolution, 4(1), 101-108.*

**Data analyses**

**The data analysis section should clarify how the data were handled and what was considered a replicate. In the methods section, the transects are referred to as replicates but the 6 plots per transects represent different temperatures. I therefore assume that the models are based on plot-level data. At the very least, transect or sampling plot should be included as a random effect in the models (ideally plot nested within transect to account for the experimental design). Including location in the models could help deal with the high variability. In addition, figure 1 shows regression lines but no regressions are described in the analyses.**

*We thank the reviewer for her input on the model regression analyses. The linear mixed model as used for analysing most of the data in the manuscript is described in the lines 205-216. We used sampling year as a random factor to account for sampling differences and interannual variabilities between the two sampling campaigns.*

    *Using plot ID as a random factor would confound the model results, as soil warming level is inherent to the plot ID, and soil warming is included as a main factor. As the reviewer suggested, in the revised manuscript we now included transect as a random factor, in combination with sampling year in a crossed random effects design. This adaptation only slightly altered the results and did not affect the conclusions of the manuscript. The statistics were edited in the figures and throughout the main text.*

**I note that these issues do not detract from the interesting and potentially important findings on the differences between soil depths and long- vs. short-term warming. However, the presentation ad discussion of the results should be revised to ensure the main messages are accurate and the limitations of the study are clear.**

**With kind regards**

**E.J. Sayer**

**Additional minor comments by line:**

**L15 – suggest replacing "lead to increased" with "increase"**

*The reviewer's suggestion was implemented.*

**L17 – what is meant by "sign"? Do you mean whether the feedback is positive or negative? This could be rephrased to make it clearer.**

*We indeed meant whether the feedback is positive or negative. The paragraph was extended, and the word 'sign' replaced with 'direction' to clarify: ' However, soil warming could also be expected to result in increased and/or unaltered SOC stocks, if, e.g., there is rapid acclimation of microbial communities or if plant productivity increases strongly. This implies that the strength and even direction of this carbon cycle-climate feedback are, highly uncertain (Crowther et al., 2016; Todd-Brown et al., 2018; van Gestel et al., 2018).'*

**L20 – omit soil before SOC**

*'Soil' was omitted.*

**L22 – extrapolations of responses from, or model parametrisation based on, short-term experiments**

*A comma was included in the sentence.*

**L27: above it**

*The suggestion was implemented.*

**L57: lose SOC (not 'loose')**

*The error was corrected.*

**Line 67 states: "Even grasslands that had been warmed at least 55 years exhibited no larger SOC loss than that observed after 5 years of soil warming." How were SOC losses over 55 years assessed without analysing samples from 55 years ago? Or does this instead mean that the SOC stocks in the long-term warmed transects were similar to those in the medium-term transects?**

*This does indeed mean that the SOC stocks in the long-term warmed transects were similar to those in the medium-term transects. Hence, the sentence was edited to match the observations exactly on line 72-73: "Even grasslands that had been warmed at least 55 years exhibited no larger SOC stocks than that observed after 5 years of soil warming."*

**L81-83: you present no data for microbial communities in this study, so upon what basis do you infer there is no evidence for physiological adaptations or compositional shifts?**

*The basis for this claim is in the next sentence. To make the meaning of the paragraph more clear, we edited the paragraph as following (ln 89-93): "Alternatively, ephemeral SOC stock loss under warming may have resulted from physiological adaptations (Allison et al., 2010; Bradford et al., 2019) or compositional shifts (Melillo et al., 2017) in the microbial community. However, previous research in these grasslands showed that soil microbial carbon use efficiency (CUE) remained constant under short- and long-term warming (Walker et al., 2018), and microbial community*

*composition was only affected by more intense (>9 ∘ C) long-term warming (Radujković et al., 2018), meaning evidence for such mechanisms is lacking."*

**L103-106: The differences to the other studies referred to here could be partly explained by differences in sampling depth, e.g. Lin et al. considered soils to 60 cm depth, Soong et al considered soils to 1-m depth and even Jia et al. considered 30-40 cm. In addition, I believe that all of these studies focussed on early or short-term changes in SOC, which were not considered for the subsoil in this work.**

*As the reviewer suggests, the short-term effect of the other studies might play a role here. However, Lin et al. and Soong et al., considered forests and Jia et al. found grass roots up to 85cm depth. Hence, all these cited studies have a deeper rooting zone than ours, which we think is a more important determinant.*

**L165 states that medium-term warmed soils were too shallow to sample deeper than 10 cm and yet lysimeters were placed at 30-40 cm depth (L177). If there were at least some sites with deeper soils in the medium-term transects, why were they not sampled?**

*We understand the concern of the reviewer. The diameter of the Prenart Super Quartz lysimeter was only 21 mm. Because of the small diameter, coring to install the device was feasible down to 30-40 cm depth in-between the rocks. This was not possible with the wider soil corer.*

**L130: incorrect spelling of Agrostis**

*We thank the reviewer for pointing out this error. It was corrected.*

**Figure 1: There seems to be a bias towards more plots at the lower end of the warming gradient, whereas at the hotter end of the gradient, it looks like there are only 3 plots – It would be useful to give an indication of the spread of the data along the warming gradient.**
        **If most soil C loss occurred during the first 5 years of warming, why does the regression include data from both sampling times? It also looks like not all plots were sampled at both times – is this correct?**

*The reviewer is right that the density of our plots decreases with soil warming (see figure below). However, since there are no assumptions in linear model about the distribution of the independent variables, this should not be a problem for our statistical analysis. The residuals of the linear mixed model met the normality and homoscedasticity requirements.*

[Figure]

*The regression
line as shown in figure 1 shows result of the linear mixed model. Hence, no separate regression
lines were shown for different grasslands (MTW vs. LTW) or sampling times, since there was no
statistically significant effect of them in our model.*

*All plots were sampled at the sampling campaigns in 2013 and 2018. Unfortunately, for
some of the warmer plots, the sample measurement failed or the data point was removed because
it was an outlier (deviating more than 3 standard deviations from the mean of the data set). This
was the case for 4 out of 122 data points.*

**Figure 2: The smoothing lines are misleading, because they imply "no change" between 10 and
50 years, for which there is no evidence**

*The reviewer is right that in our 'space-for-time' approach, we cannot infer that there would not be
any change from 10 to 50 years of warming. The smoother lines in figure 2 show that we do not
find a difference in soil warming response between 10 years of warming at the MTW grassland and
50 years of warming at the LTW grassland. As this conceptual graph is a representation of the
rationale of our manuscript, we believe it can be left in as it is.*

**Figure 3: Are these the data for topsoil C fractions? Please clarify in the legend. Are there
fractionation data for the subsoils?**

*Indeed only topsoil data was used for figure 3. This is clarified in the first sentence of the legend:
"Relative mass, soil C % and absolute soil C amount of soil aggregate fractions originating from
topsoil in the medium-term warmed (MTW) and long-term warmed (LTW) grassland."*

*Unfortunately we do not have fractionation data for the subsoils. For analysis of the
subsoil fractionation data we refer to Poeplau et al., 2017.*

*Poeplau, C., Kätterer, T., Leblans, N. I., & Sigurdsson, B. D. (2017). Sensitivity of soil carbon fractions
and their specific stabilization mechanisms to extreme soil warming in a subarctic grassland.
Global Change Biology, 23(3), 1316-1327.*
**I was attracted by the observation system set up to study the effects of global warming on SOC stocks and globally by the results obtained.**

**Beyond its originality, the observation system built offers above all a serious opportunity to study the response of C stocks in grassland soils both in the medium and long term, in a solid manner. Indeed, the observations were carried out in situ on local warming gradients applied naturally, i.e. by fully preserving the components of the ecosystems in place. This makes it possible to integrate in situ the combined responses of all these components on the soil C stock. It is rare and very valuable to have such observations.**
          **For this reason and in order to improve the readability of the document, a more explicit synthetic description of the device (distribution of the plots around the hot spots, number of Institute plots...) should be presented in the main text (from line 30). Finally, a reference to appendix A must be added in the main text.**

*We thank the reviewer for the positive feedback on our paper, and the thorough reading of the manuscript. The set-up of the two experimental sites is thoroughly described in Sigurdsson et al. (2016). As the reviewer suggested, we added a more specific reference in the introduction at line 32-33: "To address both these challenges, we determined SOC stock changes along natural geothermal gradients at the ForHot research site in Iceland, which is extensively described in Sigurdsson et al. (2016).". Also a reference to appendix A was made in the main text on line 34-35: "An elaboration on the choice of these two soil layer depths is provided in the material and methods section in appendix A."*

**I would like to emphasise that the observation set-up created by the authors shows remarkable rigour. Many precautions were taken in the design of the soil samples on transects distributed around the geothermal points. In addition, a great deal of expertise was deployed to verify and characterise the heating gradients obtained at the soil surface and at depth. The measurement methods used are not only well documented but also show a high level of rigour in the acquisition of field data. I particularly appreciated the efforts made to estimate the SOC stocks by taking into account the evolution of the apparent soil densities under the effect of warming (corrected SOC stocks).**

*We thank the reviewer for the appreciation of our work.*

**Thanks to all these investments, the results are mostly convincing. They clearly show an ephemeral SOC loss under warming from the upper soil layer. On the other hand, I am less attracted and convinced by the results and conclusions concerning the SOC dynamics in deep soils. Indeed, only soils under long-term warming could be sampled, which weakens the level of investigation on the response of deep SOC to warming. As a result, these deep SOC results and their interpretations are less robust and original than the previous ones, although 3 paragraphs are dedicated to them against only one previously (L79-83). A better balance between these two axes of results should be found, which would reinforce in this paper the place given to the**

**temporal dynamics of the SOC response to warming, for which the results obtained in this original device are clearly convincing.**

*We agree with the reviewer that the mechanism(s) behind the stable SOC stocks in subsoils is (are) not that clear as those behind the SOC stock loss in topsoil. For the latter, Walker et al. (2018) provide a well-founded mechanistic explanation, which makes a thorough elaboration on the subject unnecessary. For subsoil, there is more speculation and three (non-mutually exclusive) mechanisms were listed in the manuscript. The conclusion sentence of the subsoil paragraph (ln 114-116) emphasises this: "Further research is needed to unravel the drivers of these contrasting subsoil SOC responses to warming among experiments, which may be related to differences in soil properties, aggregate dynamics or rooting depths."*

**The ephemeral SOC loss under warming from the upper soil layer has been observed here on a natural warming device. These results are in line with other results showing an ephemeral increase of soil respiration but obtained under experimental (i.e. artificial) warming conditions. It is interesting to note that these studies, although distinct in terms of the ecosystem studied (artic grassland vs. temperate forest), share similar soil warming gradients and lead to similar durations of increase in C loss fluxes (5-6 years) before this positive response to warming disappears. Among the possible explanations already proposed in the literature to address this change in response to warming over time, several are mentioned here and sometimes tested in coherence with data collected on site when available. Nevertheless, one explanation proposed in the literature is totally absent, although it would offer an additional interpretation here that would be particularly consistent with the idea put forward by the authors regarding "the interplay between soil microbial biomass and activity" (L77). This is the positive temperature effect on enzyme inactivation which results in a reduction of the catalytic power of enzymes under warming (Alvarez et al. 2018). By accelerating the inactivation of enzymes produced by microbes, warming leads to a reduction of the enzymatic breakdown of soil organic C in the long term. Moreover, when taking into account this temperature-dependent process, which constrains the dynamics of the enzyme pool, simulation shows an ephemeral positive response of enzymatic activity, which is followed by a disappearance of the positive long-term effect of warming or even a negative effect. Considering this theory would also contribute to the discussion on the response of the soil microbial biomass (negative) and CUE reported by the authors. Finally, this theory would bring an additional element to the discussion on the carbon-climate feedbacks of warming.**

*Thank you for pointing out this interesting mechanism. It could indeed play a role in attenuating the warming-induced increase of microbial soil organic C decomposition. We included a sentence describing the mechanism on line 88-89: "Additionally, soil warming might reduce the catalytic power of microbial enzymes and lower SOC decomposition (Alvarez et al., 2018).*

**Concerning data analysis I have some comments regarding the statistical models used to analyse the effect of warming on variables. In particular, I found it difficult to digest the construction made in figure 1 and thus to adhere completely to the interpretations made in terms of warming effect. My first concern was that the data used to construct the relationship combine the two sampling dates for each field plot, i.e. two different warming durations for each field plot (confounding effects). Reading the last paragraph of Appendix A, which briefly describes the mixed model used, reassured me a bit. Nevertheless, a more explicit presentation of this model with statistical results obtained and interpretation for at least the SOC stock variable would be necessary in the appendix (with reference to appendix in the main text and the caption of Fig1).**

*We agree with the reviewer that a more explicit description of the statistical model used should be presented in the manuscript, which we now added in appendix A. We edited the mixed model, as reviewer 1 suggested, by including also plot transect as random factor. A reference to the model description in the main text and in the caption of figure 1 was also added.*

**Technical comments**

**L50 - I did not understand where the number in brackets "or 8.8 °C-1 " came from or its unit. Please clarify.**

*We thank the reviewer for pointing this out. The value between brackets provides the percentual SOC stock loss per degree Celsius, so a % sign was lacking. This error was corrected.*

**Appendix A - L163 – "we adopted a regression approach" Please specify briefly the purpose.**

*The reviewer is right that the purpose of the sentence is unclear. After consideration, we removed the whole sentence from the experimental design paragraph, as all necessary information about the statistical analysis can be found later in the appendix.*

**Appendix A - L185 – "for a some weeks" remove "a"**

*The error was corrected.*

**Fig 2a legend. SOC stocks are expressed as "kg ha-1". Replace by ton ha-1 ?**

*The error was corrected.*

**Fig 2 legend. please add 0-10 cm to the axis units**

*'Topsoil' was added to the axis units.*

**Fig 2 - It is surprising to see only 2 data (points) for the 4.1-5.1°C warming, which are also well above the curve for this warming category. Are there any missing dots or is there a problem with the legend colors?**

*The reviewer is right that this can be confusing. This conceptual figure is built as is stated in the figure legend: 'Soils are divided in four warming categories for representation. The colours on the heatmap and the smoother lines are based on a linear regression equation per sampling event.'*
        *Hence, the smoother lines are based on the linear regression of all data points, and not only on the points within a warming category. They are only drawn to visualise that no additional reduction of SOC stocks was observed after 5 years of warming, and are not intended to represent the statistical model. For this, we refer to figure 1.*

**Figure B1 – In the caption, rather than "Reduction of..." I guess that it would be more appropriate "Uncorrected carbon stock..."**

*We thank the reviewer for the suggestion. The caption was edited accordingly.*

**L70 - please move up the reference to fig3 from line 71 to line 70**

*The figure reference was moved.*

**Fig 3: The x-axis shows soil warming range from 0 to 18°C. This is different from the soil warming range 0-6.4°C. Please clarify and give explanation in the text.**

*We agree with the reviewer that using the full warming range for soil aggregates instead of soil warming <6.4°C can be confusing. However, as the aggregate data is meant to support a mechanistic theory about aggregate breakup reducing the physical protection of SOC, we do not think this is problematic. To indicate this we added the following to the figure caption: "The full soil warming range is used here, to make optimal use of the smaller sample number for aggregate data".*

**L72, 73 and elsewhere – It would be more explicit to replace "soil C %" by "C concentration" in soil fractions.**

*The suggestion of the reviewer was incorporated throughout the manuscript.*

**L70 – The name of fractions should be improved to increase direct readability without increasing words number. ">2 mm" corresponds to 2-8 mm, ">250 μm" corresponds to 2000-250 μm, ">63 μm" corresponds to 250-63 μm.**

*We thank the reviewer for the suggestion. The manuscript was edited accordingly.*

**L76 – "emerged from the interplay between soil microbial biomass and activity". Wouldn't it be clearer to talk about a change in the specific activity of the microbial biomass?**

*We thank the reviewer for the suggestion. A detailed description of the mechanism can be found in Walker et al. (2018). As not only a warming-induced increase in specific activity of the microbial biomass, but also a reduction of the microbial biomass size leads to an eventual new equilibrium of SOC stock, we decided to leave the description as it is now.*

**L77 – "Warming at the same study site accelerated microbial growth and respiration (Marañón-Jiménez et al., 2018; Walker et al., 2020)" – Please could you clarify here whether these two warming-induced accelerations were observed only in the medium term or also in the long term?**

*The sentence was extended as following (ln 83-84): "Warming at the same study site accelerated microbial growth and respiration both in the medium-term and in the long-term warmed grassland (Marañón-Jiménez et al., 2018; Walker et al., 2018)"*

**L79 – Reference to fig B1a. Add "a" and "b" in fig B1.**

*a) and b) were added to the figure panels.*

**With kind regards**

**RC3: 'Comment on bg-2021-338', Anonymous Referee #3, 02 Mar 2022 reply**

**This manuscript represents a well-planned and well thought study of soil warming on the SOC changes in both "topsoils and subsoils" in Andosols in Iceland.  The field study and sampling scheme are well organized, and methods of analysis are clearly described. The interpretations of the results are reasonable. This study would certainly contribute to global models to accurately project future SOC stocks for Andosols in similar climate regimes.**

**Thus, I would recommend thus manuscript be accepted with very minor revisions.**

**As the authors mentioned at the end of the abstract that "for this soils type and should be investigated for soils with other mineralogy".  I cannot agree anymore. But as a pedologist, I like to point out that from the information provided in the manuscript, the "topsoil" (I prefer to call it the surface horizon) is an A horizon, and the 10-30 cm is more likely a Bw horizon. In the A horizon, the dominant process is biochemical (breakdown decomposition of raw organic matter such as fine roots) whereas in the subsurface horizon is biogeochemical weathering. If this Andosol is moderately or highly weathered, then the clay mineralogy is dominated by allophanic or poorly crystallized clay minerals. This fraction usually forms strong bounding with colloidal organic compound. The authors cited Lin et al. work that was conducted in a subtropic forest. Please note that the soil profile arrangement of the grassland and the forest is different. In the forest soil, there is an organic horizon with different degree of decomposition. Then beneath the O horizon there is usually an eluvial horizon (E horizon, surface mineral horizon) and the roots in this horizon are usually dominated by coarse (big) and medium roots. The fine roots are distributed in the subsoils, or B horizon). So, the results from a grassland soil may not well applied to a forest soil. When you say "soils with other mineralogy", you actually mean non-Andosols; soils not formed from tephra. Most grass land soils in the temperate and cold temperate and subarctic regions are formed in loess with mixed mineralogy.**

*We thank the reviewer for the positive feedback on our manuscript, and for the valuable input to the discussion on soil mineralogy and pedology of the investigated soil.*

**L. 42. "The soil type on both study sites is Andosol". Suggest revising as "Soils on both study sites are classified as Andosols according to World Reference Base (WRB):**

**IUSS Working Group WRB. 2015. World Reference Base for Soil Resources 2014, update 2015 International soil classification system for naming soils and creating legends for soil maps. World Soil Resources Reports No. 106. FAO, Rome**

*We thank the reviewer for the suggestion. The citation was edited accordingly.*

**L. 258. Capitalize: Soil Survey Staff, 1999, Soil Taxonomy. Soil Taxonomy has been updated. See the 2014 version. But, where was this publication cited in the manuscript?**

*We thank the reviewer for the suggestion. The most recent version published on the USDA website (https://www.nrcs.usda.gov/wps/portal/nrcs/detail/soils/survey/class/taxonomy/?cid=nrcs142p2_053577) is the one we cited and we extracted the Andosol coverage of 0.8 % from. The most recent version of the 'Keys to soil taxonomy', which unfortunately does not indicate a coverage of Andosols, was indeed published in 2014.*

*The citation was edited according to the reviewer's suggestion and to the guidelines on the USDA website: "Soil Survey Staff: Soil Taxonomy: A Basic System of Soil Classification for Making and Interpreting Soil Surveys., Natural Resources Conservation Service. U.S. Department of Agriculture, 2nd edn., 1999."*

---

## Author Response (AR2)

Dear Dr. Fontaine

Thank you again for taking the time to evaluate our manuscript. We have incorporated the suggestions of the reviewer, and highlighted the changes in an updated version.

We hope that the changes we made addressed these comments satisfactorily.

Best regards

Niel Verbrigghe, on behalf of all co-authors

I thank the authors for their revision of the manuscript, which satisfactorily address most of the points in my previous review. I find the paper clearer and easier to read, but I remains troubled by a key point that authors did not really address in the revised manuscript and yet was my main concern.

*We again thank the reviewer for the efforts put in the review process and for the useful comments on our work.*

Indeed, I find that the presentation of the results on subsoil SOC stocks tends to overuse them to state on the stability of subsoil SOC over all the long term period of warming. Authors are aware of this point as they suggested L109 "it can also not be excluded that SOC stocks in subsoils only appear stable".
I point out again that comment that I think it is very important to avoid a too categorical and generic message on the dynamics response of deep-SOC to warming (stability) obtained from only one warming-duration (Long-term warming >50 y) and in only one area-site studied. I suggest here minor changes in the text which could attenuate such a presentation issue.

In the abstract section:
L9 – rather than "SOC reduction only occurred in topsoil" it could be "SOC reduction was only visible in topsoil"
L9-10 – "SOC stocks in subsoil (10-30cm),…, remained unaltered, even after >50 years of warming". It could be "SOC stocks in subsoil (10-30cm),…, showed apparent conservation after 50 years of warming". Note that I also propose here to remove "even" which implicitly suggests that in the short-medium terms this apparent conservation was also observed (while it has not been measured).

*We incorporated the changes on subsoil SOC stocks suggested by the reviewer in our abstract.*

In the main text section:
L97 – SubTitle – Authors could add "Apparent" to the subtitle "stable subsoil SOC stocks"
L99-100 – "We hypothesised that the similar warming intensity across the soil profile (fig. B4) would elicit similar declines in subsoil SOC stocks than those in topsoil". I suggest to precise "in the long term".
L100 – rather than "In contrast, SOC remained constant in the subsoil, even under 50 y of warming…" it could be "In contrast, SOC stocks showed apparent conservation in the subsoil under 50 y of warming…" I also propose to remove "even" here (see my comment above).

*Also here, the edits proposed by the reviewer were incorporated in the main text section.*

L100-110 – I wonder if it wouldn't be clearer to start the discussion on the lack of SOC stock reduction over the long term with the "third" point (L108-110).

*We thank the reviewer for the suggestion on the structure of the paragraph. We did not change the sequence of the reasoning here, because the absence of increased dissolved organic C (DOC) with warming in subsoil makes the mechanism of compensated subsoil SOC stocks by C influx from topsoil unlikely.*

Another minor comment:

**L168 (Appendix A) – Authors mention that "all measurement plots had similar soil depth" while L175 indicates that the MT grasslands soils were too shallow...**

*Also this last comment was changed according to the reviewer's suggestion. We thank the reviewer for the careful reading of our manuscript.*